# Community Readiness in Implementing Sustainable Tourism on Small Islands: Evidence from Lombok, Indonesia

Rosiady Husaenie Sayuti

Department of Sociology, University of Mataram, Jl. Majapahit 62, Mataram 83125, Indonesia; sayuti@unram.ac.id

**Abstract:** Implementing sustainable tourism development is an essential part of the strategy to achieve the 2030 Sustainable Development Goals and inclusive, sustainable economic growth. This research aimed to examine the level of readiness of the community to carry out sustainable tourism development, especially community-based tourism on small islands, such as Lombok Island, Indonesia. More specifically, the objectives of this study were as follows: first, knowing the level of community readiness in carrying out sustainable tourism development; second, knowing the various dimensions that influence sustainable tourism development; third, understanding the role of stakeholders in sustainable tourism development. Furthermore, the role of the education sector in increasing community readiness for sustainable tourism development is an exciting matter to study. The method applied in this research used mixed techniques, a combination of quantitative and qualitative approaches. Quantitative methods were used to determine the level of community readiness in sustainable tourism development. A qualitative approach was used to determine the various factors that influence the development of community-based tourism and to determine the role of stakeholders in the development of community-based tourism. Community readiness for sustainable tourism development is high in terms of economic, social, cultural, environmental, and symbolic capital aspects. Economic, social, cultural, and natural elements influence sustainable tourism development. Stakeholders who play a role in developing sustainable tourism in the research area are the government, the private sector, universities, non-governmental organizations, and the media. The results of this study can be used to create a government policy related to sustainable tourism development.

**Keywords:** sustainable tourism; stakeholders' role; community preparedness; Lombok; Indonesia

## 1. Introduction

World leaders agreed to include tourism development in the 2030 Sustainable Development Goals agenda, as contained in the goals related to inclusive and sustainable economic growth, responsible consumption, and the conservation of life in the waters (Goals 8, 12, and 14). This goal is justified because tourism development has grown into a sector with great potential. In recent decades, tourism has become the sector with the highest growth [1,2]. However, in its development, the tourism sector is inseparable from various weaknesses and shortcomings and has even had an unexpected impact. There is very little evidence that tourism can reduce poverty, especially in pockets of poverty [3]. One of the factors suspected to be the cause is the unequal relationship between tourism actors in the village and investors and other tourism actors from outside. Inequality between economic and social capital is common [4]. The 2022 Sustainable Development Report by the United Nations stated that Indonesia's achievement of the SDGs is 69.16%. However, compared to other countries in Southeast Asia, Indonesia ranks fifth, below Thailand, Vietnam, Singapore, and Malaysia. This information means there are still many programs that must be improved so that Indonesia can achieve the SDGs by 2030, according to the targets that have been set. Sustainable tourism development is included as part of the program in the SDGs. According to several researchers [5–8], this imbalance problem can be overcome

through education, namely by increasing people's access, including that of people with low incomes, and by aligning the learning curricula of educational institutions more closely with the principles of sustainable tourism development. However, according to Alam [5], such an imbalance is also a problem in the world of education, which must be addressed first. Therefore, one of the suggestions is to apply the principles of equity and sustainability in educational institutions so that people with low incomes have the same access as other citizens and communities to opportunities for education and higher education.

UNWTO estimates that by 2030, the number of travelers worldwide will reach 1.8 billion people [9,10]. Various anticipatory programs need to be encouraged. If not, damage in various fields caused by tourism will be unavoidable, such as damage to natural resources and the environment, economic loss, degradation of cultural values, and even loss of cultural heritage and its uniqueness [10,11]. It is the background for the growth of the idea of developing eco-friendly tourism, better known as sustainable tourism. In UNWTO's view, what is meant by sustainable tourism is a tourism activity that still pays attention to aspects of social, environmental, and economic sustainability both for the present and for the future [12]. The goal of sustainable tourism is to improve the short and long-term welfare of the surrounding community, meet the demands of visitors, and at the same time, protect or maintain the environment. According to A. Guizzardi et al. [13], the primary indicator of the success of sustainable tourism development is the preservation of natural, environmental, and cultural resources. In addition, tourism has a comparative advantage that can trigger economic growth in an area.

The problem is that precisely applying sustainable tourism is not easy. As a result, failures in tourism development based on sustainability, including, in this case, environment- and culture-based failures, often occur [14,15]. Moreover, the development of tourism in various places causes damage to the environment and even culture. Therefore, it is essential to have good planning that prioritizes aspects of sustainability [16].

S. Guo et al. [17] stated that sustainable tourism development faces many challenges, including infrastructure, processes, procedures, and the equipment needed. Building a balance between the environment, society, and economy is not easy. Tourism development is very dynamic and continues to experience changes in developing and developed countries [18]. In later developments, various creations grew in the tourism community to anticipate different challenges and failures in sustainable tourism development. One is the emergence of the concept of community-based tourism development. Implementing this concept is expected to prevent or reduce various negative impacts of tourism activities [19]. Community-based tourism essentially manages tourism from the community for the community. At the same time, it educates tourists to love the environment and not damage the various social and cultural attractions in tourist destinations [20]. The World Wide Fund for Nature (WWF) defines community-based tourism as the mastery and management of tourism by the local community; most of the benefits obtained are also enjoyed by the local community [21].

Lombok Island is an island whose tourism has only begun developing in Indonesia. Its location close to the island of Bali gives this island very high value and potential for development. The government designated the Mandalika Area on Lombok Island as a National Super Priority Area. Now, the international Moto GP event has begun to be held in this area and is predicted to bring in more than 100,000 tourists every time the Moto GP is held. Lombok Island is also known as a Halal Tourism Area. On 20 October 2015, the World Halal Travel Summit in Abu Dhabi, United Arab Emirates, awarded Lombok two designations at once, namely as the World's Best Halal Tourism Destination and the World's Best Halal Honeymoon Tourism Destination [22].

The Provincial Government of West Nusa Tenggara has designated 99 villages as Tourism Villages. This policy aims to develop rural tourism to prepare West Nusa Tenggara, particularly Lombok Island, as a national priority tourist destination. Three of the 99 Tourism Villages are located in Jerowaru District, where this research was conducted. The three

villages are Sekaroh, Jerowaru, and Seriwe. Most of these villages have high potential in the tourism sector but are still in the preparation stage, still developing tourist destinations.

### 1.1. Research Purposes

Tourism studies on these small islands are still attractive, considering several factors, first, in terms of the characteristics of small islands that are vulnerable to disasters but rely on natural tourism as the primary source of livelihood for the population. Second, it is essential to pay attention to the population's readiness level for tourism development, primarily community-based tourism, to anticipate various negative impacts of tourism development. Understanding this readiness level is very important to design a community-based tourism development framework in the future. In this study, readiness means people's knowledge of and behavior toward community-based tourism concepts and principles. Third, research on the locus of small islands is still very significant, considering that Indonesia has more than 15,000 small islands stretching from Sabang to Merauke, from the western tip to the eastern end of the archipelago, with various social and cultural characteristics, and their respective natural environments.

Although research on community-based tourism has been conducted in various parts of the world, very few researchers have made small islands their locus of study; among the few are the study by Giampiccoli and O. Mtapuri [23]. It is stated that tourism on small islands is like a double-edged sword. On the one hand, it can increase economic growth, but also, on the other hand, it can reduce and even destroy the community's economy [24]. Furthermore, it is explained that the challenges that become issues in small islands can be classified into four categories: the small island size, remote location, vulnerable environment, and other socio-economic factors. In addition, many of these islands have limited natural resources, and their dependence on external resources is very high [25]. Other concerns of academics regarding these small islands are their carrying capacity, community participation, and environmental policies [23,26].

Globally, there are around 180,000 islands of various sizes and tourism potential. However, only about 90 islands are larger than 10,000 square kilometers. The rest are small islands with unique characteristics, which are outermost, left behind, or even have exotic characteristics as attractive tourist destinations [25,27]. Few small islands in the world rely on tourism as the backbone of their economy, including in the context of poverty alleviation.

Therefore, based on the description above, it can be explained that this research is fundamental to assessing community readiness for sustainable tourism development on small islands, such as Lombok Island, Indonesia. Knowing the level of preparedness of the community to carry out sustainable tourism development is essential to avoid the possibility of negative impacts from tourism, as happened in other places. Examining how education plays a role in increasing community readiness for sustainable tourism development is no less critical.

More specifically, the objectives of this study are as follows: (1) to determine the level of community readiness in developing sustainable tourism; (2) knowing the various factors, including education, that influence sustainable tourism development; and (3) knowing stakeholders' role in sustainable tourism development.

### 1.2. Literature Review

Community-based tourism development is a strategic instrument as a means of development. To examine this matter more deeply, we must explore the extent to which tourism is a determining factor in social and economic change and even culture. According to several researchers, M. del Río-Rama et al. [21], M. Cawley et al. [28], and F. Zach and P. Racherla [29], one of the factors that also influences tourism development is the network factor. Therefore, tourism actors should be able to build a network outwardly with fellow tour operators and non-governmental organizations committed to tourism development.

According to S. Asker et al. [30], and N. B. Salazar [31], community-based tourism (CBT) development has been developed since the 1990s to increase local community partici-

pation in tourism development. In addition, through expanding the role of the community, it is hoped that the concept of sustainable development will be more effectively implemented so that not only will economic benefits be obtained, but also, at the same time, there will be the preservation of natural resources and cultural values that are the treasures of the local community. Giampiccoli and O. Mtapuri [23] highlight the development of conventional tourism compared to community-based tourism in Vietnam. According to them, there must be a balance between the two types of tourism. With community-based tourism, it is hoped that it will involve more micro, small, and medium enterprises (MSME) actors in the accommodation and food sectors. This is in line with government policies related to economic openness, empowerment, and poverty alleviation [1,32].

V. Rachmawati et al. [33] mentioned that the CBT concept could be an effective model for providing benefits to local communities. S. Nitikasetsoontorn [34] explains that seven factors are the key to the success of CBT: community participation in decision-making processes, local ownership, collective responsibility, leadership, governance, uniqueness, and the ability to be different. In the view of E. Ruiz-Ballesteros [35], CBT is an intelligent strategy to build and increase community resilience in realizing social and environmental sustainability. Research by Ballesteros and Tejedor examines the interrelationships between CBT implementation and community resilience in social ecosystems. Their research also makes small islands their locus, namely the island of Floreana in the Galapagos archipelago.

In the opinion of T. Dangi and T. Jamal [36] and E. So and R. Spence [37], CBT is expected to increase local community participation, improve management and service quality in the long term and bring benefits to the community in the field. In other words, CBT is expected to implement the principles of sustainable development and community development in general. However, in the implementation process, management by local communities requires work. Especially in suburban areas, help and support from outsiders are still needed [38].

## 2. Materials and Methods

This study used a mixed-methods approach. According to P. Clark [39] research with a mixed-methods approach has its philosophical assumptions and information-seeking framework. Moreover, this method has clear instructions for collecting and analyzing data from various sources. According to [40], mixed methods have multiple advantages in solving complex problems because they integrate the philosophical frameworks of post-positivism and interpretivism, aligning qualitative and quantitative data to be explained meaningfully.

Mixed methods also offer flexible logic and methodologies and a deeper understanding of a simple study [41]. Other researchers, G. Enosh et al. [42] and S. Dawadi [43], stated that with mixed methods, researchers could explain answers to their research questions quite broadly and in-depth. Through a quantitative approach, extensive information can be obtained and used to generalize a situation. News or issues can be explored in more depth using a qualitative approach. In the context of this research, a quantitative approach will be implemented to obtain data to answer objective 1: knowing the level of community readiness for sustainable tourism development. Meanwhile, a qualitative approach is used to obtain data and answer objective 2, namely, to identify the various dimensions that influence sustainable tourism development, and objective 3, to know the role of stakeholders in sustainable tourism development.

### 2.1. Data Collection Technique

Data collection methods used in this study were as follows. (1) Observation was used to directly observe the activities carried out by the research object closely. (2) A questionnaire was used as a data collection technique that gave the respondents a set of questions or written statements to answer. This questionnaire was used for the process of collecting quantitative data. The respondents selected to answer the questionnaire questions were community members involved in sustainable tourism activities in the research area.

The data were processed by interpreting them in the form of numbers with the help of SPSS 16.0, making it easier for researchers to analyze the data obtained. (3) Documentation was used to obtain data and information in the form of books, archives, documents, written numbers, and pictures in the form of reports and information that could support the research. Documentation was used to collect data and then reviewed. These data were collected through various sources of written data related to objective issues and other supporting data. (4) To obtain qualitative data, in-depth interviews were held with selected key informants using the following considerations [44]: (a) informants have authority and know and understand tourism activities around the location; (b) informants carry out activities within the scope of the tourism area; (c) informants have enough time to be asked for information; (d) informants who are not in a state of illness or experiencing other health problems; (e) informants are considered to understand the tourism development under study.

Seven key informants were selected in this study: the village head, the Forest Farmers' Group (KTH) head, the hamlet head, the parking attendant, one tourism manager, and two traders. Then, the data collected were analyzed based on the stages of analysis referring to Miles and Huberman [45], namely reduction, presentation, and conclusion.

### 2.2. Research Sample

This research was conducted in Jerowaru District. The villages used as research locations were Sekaroh, Jerowaru, and Seriwe. The consideration was that in Jerowaru District, only three villages have been designated as Tourist Villages based on West Nusa Tenggara Governor Decree No. 050.13-366 the Year 2019. For quantitative data collection, the sampling technique was used. The sampling technique used was quota sampling. Quota sampling is a sampling method in the category of non-probability sampling, which does not provide equal opportunities for each element or member of the population to be selected. Quota sampling is a technique that first determines the number and specific characteristics of targets that must be met [44]. The features of the samples taken in this study were those who are directly or indirectly involved in tourism activities. Because this research was a type of correlational research, the sample size comprised 60 people. According to Sugiyono [44], the number of appropriate samples for a correlational study is in the range of 30 to 500. The consideration for using quota sampling was that the number of those directly involved in tourism activities could not be known with certainty. Meanwhile, the accidental sampling approach was used to select respondents, namely interviewing those found at the research location who were considered to meet the requirements for respondents.

### 2.3. Data Analysis

To measure community readiness in community-based tourism development or sustainable tourism, we used four variables: economic capital, cultural capital, social capital, and symbolic capital. Indicators for each variable can be seen in Table 1 below.

**Table 1.** Level of Community Readiness.

| Variable | Indicator |
| --- | --- |
| Economic capital | Banking transaction tools |
| | Media to obtain information |
| | Trends in using mobile phone |
| | Mobile phone ownership |
| | Ability to use mobile phone |
| | Availability of houses as lodging |
| | Planned tourism facilities |
| | Owned tourism facilities |

**Table 1.** *Cont.*

| Variable | Indicator |
|---|---|
| Cultural capital | Opinions about education |
| | Support for education |
| | Knowledge of the Jerowaru tourist village |
| | Part of Pokdarwis |
| | Experience working in the tourism sector |
| | Foreign language skills |
| | Interaction ability |
| Social capital | Responses when there are tourists |
| | The intensity of visiting neighbors |
| | Participation in community events (customs/traditions) |
| | Involvement in promoting tourism |
| | Acceptance of cooperation with external parties |
| | Obedience to village rules |
| | Upholding local values |
| Symbolic capital | People who are trusted to solve problems |
| | Deliberating and decision-making figures in the village |
| | The need for a special tourist rule system |
| | Participation of government officials in community activities |
| | Participation in community activities |
| | The ability of leaders to lead democratically |

Resources: Purwanto and Sulistyastuti [46].

The questions were developed based on the indicators in Table 1 above. For instance, we asked whether respondents had a bank account and could use mobile phones for economic capital. We also asked them whether their houses were available for tourism, etc. For social capital, we asked respondents about their social relations, response when there are tourists, etc. Each question had alternative answers, "Yes" or "No". Then, the percentages of respondents that answered "Yes" or "No" to each question were calculated. From the accumulated answers, conclusions were then drawn, as shown in the following table (Table 2).

**Table 2.** Community Readiness Level Categories.

| Percentage (%) | Category |
|---|---|
| 11–41 | Low |
| 41–71 | High |
| 71–100 | Very high |

The categories above were obtained with the highest percentage value minus the lowest percentage value, namely 11%, and the result was 89%; then, the interval distance was found by dividing 89/3 = 29.6, which was rounded up to 30. The result yielded the categories shown above.

As for qualitative data, the data analysis process was carried out through three parallel activity flows [41]. First, data reduction, which is defined as a selection process focusing on simplification, abstracting, and transforming "rough" data that emerges from written records in the field, was performed. The data obtained in the area were extensive; for this

reason, it was necessary to record them carefully and in detail. As stated, the longer a researcher is in the field, the more complex and complicated the data. For this reason, it is necessary to conduct data analysis through data reduction immediately. Reducing data means summarizing, choosing the main things, focusing on the essential items, and looking for themes and patterns. Thus, the reduced data provide a clear picture and make it easier for researchers to collect further data and look for them when needed. The second activity was the presentation of data. After the data were reduced, the next step was displaying the data. For example, in qualitative research, data presentation can be achieved with brief descriptions, charts, relationships between categories, and the like. The third activity involved reaching conclusions. Initial conclusions are still temporary and will change if no substantial evidence is found to support the next data collection stage. However, if the collection of data is pursued at an early stage, conclusions can be formulated.

Informants are research subjects who can provide information about the phenomena or issues raised in the research. To select informants, researchers used a purposive technique. According to Sugiyono [44], the purposive sampling technique is to choose informants with in-depth information and knowledge about the aspects of the data sought. In this study, the researchers classified the informants as follows: First were key informants. Key informants know in depth the issues being researched and understand various matters related to the background and implemented policies. In this case, the key informants were the head of Sekaroh Village and the head of the Pink Lestari Forest Farmers Group. Second were main informants. A main informant is an informant who knows in depth the problem being researched and is directly involved in the activity that is the object of research. The main informants included managers of tourism activities in the Pink Beach tourist area. Third were supporting informants. Supporting informants are people who interact directly with the environment around tourist objects in the research area. Supporting informants in this study were traders, parking/ticket attendants, and the community around the research area. The researchers documented the list of informants as part of the research report, but the list is not shown in this paper.

## 3. Results

### 3.1. Characteristics of Respondents

An explanation of the characteristics of respondents can be seen in Table 3. Based on the distribution of respondents based on gender, it can be seen that the ratio of male and female respondents was balanced (50:50)%. Based on marital status, of the 60 respondents, 95% of respondents were married, and the rest were single, widowers or widows.

**Table 3.** Characteristics of Respondents.

| Characteristics | Total | Percentage |
|---|---|---|
| Gender | | |
| - Male | 30 | 50 |
| - Female | 30 | 50 |
| Marital status | | |
| - Never married | 1 | 1.7 |
| - Married | 57 | 95 |
| - Divorced | 1 | 1.7 |
| - Widow/Widower | 1 | 1.7 |
| Age (years) | | |
| - 15–25 | 4 | 6.7 |
| - 26–35 | 8 | 13.3 |
| - 36–45 | 27 | 45 |
| - 46–55 | 15 | 25 |
| - >55 | 6 | 10 |

**Table 3.** *Cont.*

| Characteristics | Total | Percentage |
|---|---|---|
| Education | | |
| - Never attended school | 1 | 1.7 |
| - Not finished elementary school | 2 | 3.3 |
| - Finished elementary school | 14 | 23.3 |
| - Not finished junior high school | 0 | 0 |
| - Finished junior high school | 17 | 28.3 |
| - Not finished senior high school | 3 | 5 |
| - Finished senior high school | 23 | 38.3 |
| - One-year diploma | 0 | 0 |
| - Two-year diploma | 0 | 0 |
| - Three-year diploma | 0 | 0 |
| - University graduate | 0 | 0 |
| Average income (IDR) | | |
| - <500,000 | 38 | 63.3 |
| - 500,000–1,500,000 | 22 | 36.7 |
| - 1,500,000–2,500,000 | 0 | 0 |
| - 2,500,000–3,500,000 | 0 | 0 |
| - >3,500,000 | 0 | 0 |

Based on age, the distribution of respondents included 45% aged 36–45 years, 25% aged 46–55 years, 13.3% aged 26–35 years, 10% aged >55 years, and only 6.7% aged 15–25 years. Most of them did not finish High School in terms of educational level. Only 28.3% finished their Junior High School. Based on average income, 63.3% of respondents earned below IDR 500,000 per month. At the same time, 36.7% had an income of around IDR 500,000–1,500,000 per month.

*3.2. Analysis*

Community Readiness Level

From the results of the research, data related to the level of readiness of the community in carrying out sustainable tourism development were obtained, as shown in the following table.

Table 4 below shows the level of community readiness in carrying out sustainable tourism development for each variable measured in this study, where economic capital and social capital were in the very high category, while cultural capital and symbolic capital were in the high category.

**Table 4.** Results of Level of Community Readiness.

| Variable | Indicator | % | Mean | Category |
|---|---|---|---|---|
| Economic capital | Banking transaction tools | 100 | 77% | Very high |
| | Media to obtain information | 100 | | |
| | Trends in using mobile phones | 98 | | |
| | Mobile phone ownership | 90 | | |
| | Ability to use mobile phone | 90 | | |
| | Availability of houses as lodging | 65 | | |
| | Planned tourism facilities | 51 | | |
| | Owned tourism facilities | 20 | | |

**Table 4.** *Cont.*

| Variable | Indicator | % | Mean | Category |
|---|---|---|---|---|
| Cultural capital | Opinions about education | 93% | 52% | High |
| | Support for education | 93% | | |
| | Knowledge of the Jerowaru tourist village | 94% | | |
| | Part of *Pokdarwis* | 11% | | |
| | Experience working in the tourism sector | 14% | | |
| | Foreign language skills | 26% | | |
| | Interaction ability | 34% | | |
| Social capital | Responses when there are tourists | 87% | 79% | Very high |
| | The intensity of visiting neighbors | 83% | | |
| | Participation in community events (customs/traditions) | 76% | | |
| | Involvement in promoting tourism | 43% | | |
| | Acceptance of cooperation with external parties | 79% | | |
| | Obedience to village rules | 87% | | |
| | Upholding local values | 90% | | |
| Symbolic capital | Honor and fame | | 60% | High |
| | People who are trusted to solve problems | 74% | | |
| | Deliberating and decision-making figures in the village | 57% | | |
| | The need for a special tourist rule system | 57% | | |
| | Participation of government officials in community activities | 57% | | |
| | Participation in community activities | 57% | | |
| | The ability of leaders to lead democratically | 59% | | |

Results of the study (2023).

1. Economic capital

Based on the results of the analysis, as shown in Table 4 above, it can be seen that the level of readiness of the community in the research area to carry out community-based tourism development or sustainable tourism in terms of economic capital variables was included in the very high category with a rate of 77%. It indicates that the people in the research area are economically ready to implement community-based or sustainable tourism.

Economic capital consists of two indicators, namely, facilities and infrastructure. Facilities are one of the things that must be considered in tourism development. Few villages in Indonesia have tourism potential but find it challenging to develop because adequate facilities and infrastructure do not support them. From the existing data, the facilities for tourism in the research area were reasonably adequate. Tourism facilities owned independently by the community included one hotel/inn, four restaurants, one snorkeling/diving equipment rental business, and 14 others (sea transportation such as boats), which are used for tourism and visits to the islands around the pink beach. Meanwhile, several community homes could be used as lodging for tourists. This is a form of community participation in sustainable tourism management that can benefit the community.

Tourism infrastructure includes natural and human resources that tourists need. Infrastructure in tourism is as critical as facilities, so these two things must be available and will determine the sustainability of tourism in the future. Based on the data analysis from this study, the infrastructure was in the category of adequate access to the village, allowing

easy passage using both land and sea transportation. Meanwhile, the latest information or news was available on TV, social media, and the internet. This was consistent with the research results where 54.2% of respondents had the latest information and news through TV, and the rest through social media and the internet. Water sources in the study area were also diverse, but some people struggled to obtain a clean water supply. The research area is a dry land with minimal fresh water sources.

2. Cultural capital

Next was cultural capital, where from the results of this study, the mean percentage of the indicators was 52%, which placed t it in the high category. In contrast to the "very high" category, this category indicated that the level of community knowledge was high, but the level of skills was relatively low. There were two indicators of cultural capital, namely knowledge and skills. These indicated that the levels of knowledge and skills of the community influence whether the community in the study area is ready or not to carry out sustainable tourism development. Therefore, the community, and especially the tourism community, must possess knowledge and skills, because these are very important for the sustainability of tourism.

The people in the research area generally understood the importance of education. From the data obtained from this study, 100% of respondents considered education essential and supported their children in the pursuit of education. This indicated that the respondents fully supported their children's education and pursuit of the highest possible level of education, with the hope that their children will have better jobs than their parents. Thus, cross-generational social mobility will likely develop with better livelihoods for the next generation. Knowledge was also related to local community knowledge about tourism development in the research area. Knowledge indicators can reflect the awareness of the community itself or the level of outreach from the village government to the community regarding community-based tourism development. Based on the research results, 95% of respondents knew that the government had designated the research location to become a tourist village for sustainable tourism.

Community skills in the research area can be viewed in terms of three sub-indicators: experience working in the tourism sector, foreign language skills, and the ability to interact with tourists. Based on the research results, 14% of respondents had work experience in the tourism sector. However, this showed that there are still relatively few people who have skills in the field of tourism, so there is a need for training to increase people's insights and skills related to tourism and the tourist village itself.

In terms of ability, in this case, foreign language skills and the ability to interact with tourists still need to be improved. However, based on the study's results, it was found that as many as 26% of respondents could speak a foreign language, and 34% had experience interacting with tourists. Community education also influences community readiness because the higher the level of a community's education, the better its citizens understand tourism and are more likely to play a role in providing creative and up-to-date ideas for tourism sustainability. Therefore, although from the data obtained, there were differences in the indicators of knowledge and skills, in general, it could be concluded that in terms of cultural capital, the level of community readiness to implement sustainable tourism was high.

3. Social capital

Regarding social capital, the analysis results showed that this variable was included in the very high category with a mean rate of 79%. This meant that the people in the research area, from the perspective of social capital, were ready to implement the principles of community-based tourism toward sustainable tourism. There were three indicators of social capital: values/norms, social relations, and trust. The findings indicated that the levels of values/norms, social relations, and trust in the research area between communities, communities and government, and communities and tourists can influence the level of community readiness to carry out sustainable tourism development.

Values and norms must exist and be adhered to by society, as they strongly influence the relationships and interactions between citizens. Without explicit norms and sanctions, relations between people tend to degenerate into chaos. This phenomenon can result in anomie, where individuals in a society tend to do whatever they want. It also applies to tourism activities. As experienced in other areas, tourism can hurt the economic and social life of a community. In addition, it is feared that it will bring about a shift in existing local values and norms due to the entry of tourists into the village with their own culture (especially Western culture), especially in terms of dress and other behaviors. Therefore, with the existing rules and values in society, the tendency for these negative impacts to occur can be prevented.

Data from this study showed that 95% of respondents felt it was necessary to prepare a system or special rules for visiting tourists to limit their behavior so that local community values can be maintained and are not affected by tourism activities. Besides that, tourists and local people are expected to respect each other. In another part of this study (Table 4), 90% of respondents answered that the people in the research area still upheld existing local values by participating in local traditions (76%) passed down from generation to generation. Communities that do not comply with existing local values will be subject to social sanctions in the form of ridicule from other communities and even ostracism. Regarding compliance with existing regulations, 100% of respondents replied that they complied. They believe that the village government, as the body chosen to lead the community, should be able to prevent things that are not desirable and contrary to the values and norms of local society. Various regulations that have been instituted are believed to be able to prevent chaos and bring order.

Social relations are an essential form of social capital the tourism community owns. Social connections are marked by a typology with group characteristics and orientation. Social groups are usually traditionally formed based on lineage and similarities in beliefs. Social relations between tourism actors must be good so that tourism activities can run smoothly and increase mutual trust between tourism actors. Based on the study results, 100% of respondents helped and attended when there were celebrations, circumcisions, weddings, and other social activities—the community benefits in the form of money, food, and labor. The social relations of the community in the research area and the culture of gotong-royong have always been powerful. This was shown from the respondents' answers to questions about visits with neighbors, where 83% stated that they often visit neighbors' homes, and even 18.6% stated that they very often see neighbors to sit or eat together.

From the research data related to responses when meeting tourists who visit Jerowaru, as many as 96.7% of respondents answered that they were friendly toward tourists with greetings and smiles or even by waving to them. This shows that limited foreign language skills did not cause people in the study area to behave poorly toward tourists. Community involvement in promoting their location as a tourist destination still needs to be improved. However, based on the research results, 43% of respondents were involved in promoting tourism. Participation from the community, especially by youth and the millennial generation, is needed to help in promotional activities, primarily through social media, which is currently very popular among youth.

Communities in the research area trusted the village government to manage tourism and the proceeds from tourism activities. BUMDES (enterprise institution owned by the village) handles the management and distribution of tourism proceeds with a stipulation of the amount of distribution based on mutual agreement. Regarding cooperation with external parties, the community in the research area was very open. This was confirmed by the research results, where 85% of respondents agreed to accept cooperation with outsiders.

Based on the discussion above, it can be concluded that in terms of social capital, from the indicators of values/norms, social relations, and public trust in the research area, respondents were ready to develop sustainable tourism. The local values of the people in the research area were positive and very much needed in tourism activities. Likewise, some

norms or rules were used to regulate the behavior of people and tourists so that community members can control the course of tourism activities.

Apart from that, the indicators of social relations between the community and other communities, the community and the government, and the community and tourists/outsiders were excellent. Social relations can be used as social capital to help smooth tourism activities. Good social relations can also generate good trust and contribute to the smooth running of activities and the sustainability of tourism.

4.  Symbolic capital

Finally, regarding symbolic capital, the mean value of the indicators was 60%, placing it in the high category. This shows that the influence of leadership, honor, and fame on community responses is dynamic. Symbolic capital had three indicators, namely leadership, integrity, and fame. These showed that the form of one's leadership, people who are respected, and people who are well-known in the community can influence whether or not the community is ready to implement a development program.

Based on the results of the study, 59% of respondents answered that the leaders in the research area could lead democratically. This finding showed that the performance of village leaders, in this case, the village head, was quite good in the eyes of the community. Another research result that supported this statement was the response by 57% of respondents that the village government was involved in community activities. Again, this showed that the community viewed government leadership in the research area as good and in line with community expectations.

Honor and fame are related to the figures most respected and known by the public. The results showed that 86.7% of the respondents answered that the village head was the most dominant decision-maker. The village head was also a trusted figure in solving community problems. This phenomenon confirms the results of the research, where 60% of respondents answered that the village head is a figure trusted by the community to solve problems. Based on the description above, a conclusion can be drawn that how a leader leads, people who are respected, and people who are famous/famous in the community, in this case, the village head, can influence the response or responses of the community regarding development. Therefore, there is a role for suggestions and the charisma of the village head in ensuring that the community will always follow and support what is considered good and right by village leaders.

### 3.3. Influential Aspects

To understand more deeply the level of community readiness for sustainable tourism development, five essential aspects are discussed below. These four aspects are economic, social, cultural, environmental, and political. The relationships of these aspects were investigated by interviewing several informants using a qualitative approach.

1.  Economic Aspect, consisting of three indicators

    a.  Funds for development

    Information obtained from informants indicated that for sustainable tourism development, the sources of funds were the village government and the Forest Farmers Group, which manages the tourism in that area. This information was revealed from an interview with the village head and the head of KTH Pink Lestari during in-depth discussions on 8 October 2022.

    b.  Creating jobs in the tourism sector

    One of the principles of sustainable tourism is being able to create jobs as well as generate income for residents. In other words, tourism development can create jobs for the surrounding community. This information was confirmed by the head of KTH Pink Lestari and another informant in the study location.

2. Social Aspect

   a. Improved quality of life

   As with changes in the economic sector, tourism also brings social and economic changes. Our informants in the study area mentioned that tourism increases people's income and quality of life.

   b. Increased community pride

   An increase in community pride or community acceptance of tourists visiting tourist attractions was expressed by an informant. The informant stated that the community in the study area is quite open to visitors. People feel pride in their village, which has been a tourism destination in the region.

   c. Building stronger community organizations

   They are strengthening community organizations in tourism areas tasked with managing and developing tourism potential so that dreams can be realized. This willingness requires supporting the community itself. Therefore, according to the informant, local tourism institutions such as the KTH organization are needed.

3. Cultural Aspect

   a. Encouraging people to respect different cultures

   At a tourist location, tourists usually bring their own culture from one place to another, including in existing tourist attractions. However, the most important thing is how the community respects these cultures. This phenomenon was expressed by an informant who acknowledged the cultural variety brought by the visitors. He said that outside cultures must be appreciated by the community, and at the same time, it must maintain its own.

   b. Facilitating the development of cultural exchange

   In the tourism area of Sekaroh Village, tourists bring their own culture. Therefore, with the presence of local and foreign tourists, cultural exchanges often occur, as stated by one of the informants.

4. Environmental Aspects

   a. Carrying capacity

   The carrying capacity, or the maximum capacity for visitors to the area, needs to be understood as one of the principles of implementing sustainable tourism. As one of the informants explained, since the tourism destination is a conservation area, people must pay attention to the carrying capacity of the region. This is one of the principles of sustainable tourism development.

   b. Raising awareness of the need for conservation

   Increasing awareness of the need for conservation around tourist areas is essential to maintain natural ecosystems. It is said that this understanding is not only for the local community itself but also for the visitors.

*3.4. The Influence of Education on Sustainable Tourism*

Curriculum development must be based on cultural values in society, in addition to knowledge, technology, politics, and the economy [47]. Furthermore, Print [48] states that culture is also a target to be achieved in implementing curriculum development. From this view, it can be concluded that the education sector is not only an instrument in sustainable tourism development but also a target that will be affected when tourism develops.

Research by Saliman et al. [49] concluded that the development of tourist villages in the research area positively impacted the development of local culture and the world of education. To minimize the adverse effects that can occur in tourist areas, it is advisable to apply the principles of sustainable tourism development. One of the principles of the

sustainable tourism program is the availability of an environmentally conscious education mechanism for visitors and the local community. This is stated in the Regulation of the Minister of Tourism of the Republic of Indonesia Number 14 of 2016, which regulates the arrangement of sustainable tourism destinations [50,51].

Zeki Akinci et al. [52], in their research on students of the Department of Tourism at Akdeniz University, concluded that higher education has a positive effect on the development of behavioral attitudes of students toward the world of tourism. In the context of the role of the curriculum in educational institutions, this study examined the extent to which the educational process has succeeded in changing student expectations, perceptions, and satisfaction during study in the tourism department. It was found that the students felt that the educational process they experienced fulfilled their expectations toward tourism.

According to Amoah and Baum [53], the process of continuing tourism education in Turkey has been augmented with various technological and infrastructure tools needed to develop modern and sustainable tourism. Furthermore, increasing human resources capacity is also continuously carried out in addition to issuing regulations related to restrictions on foreign workers in the tourism sector.

Idajati et al. [54] examined the factors influencing the form and level of community participation in developing geo-tourism in Wonocolo Village, Bojonegoro Regency. From their research, they concluded that the educational variable for visitors is at the functional level, together with infrastructure and local government policies. At the same time, the education variable for local communities is at the incentive level or below the functional level. This research shows how vital the role of education is in developing sustainable tourism.

In their research on turtle conservation in Bali, Nurhayati et al. [55] analyzed the extent to which turtle conservation education programs correlate with sustainable marine tourism in Bali. This study concluded that the respondents' education levels significantly influenced their awareness of the importance of turtle conservation in the study area. In other words, the higher a person's education level, the better their understanding of the importance of conservation activities for preserving natural resources. Furthermore, it was noted that environmental education programs are still needed. Therefore, ecological education programs must be able to be implemented on an ongoing basis, either by integrating them with the existing formal education curriculum or through informal education, such as short courses or outreach in existing tourist areas [56].

The research reports described above show that the education sector plays a significant role in sustainable tourism development.

### 3.5. The Role of Stakeholders in Tourism Development

The role of stakeholders is needed in efforts to develop tourism in Sekaroh Village. If the development program is well designed, it can potentially increase regional income. Stakeholders in the ecotourism sector include anyone who influences and is influenced by tourism development. They are residents, the government, community groups, the private sector, tourists, and other parties not directly related to tourism development. From the results of observations in the field and various forms of information obtained as part of this research, the stakeholders involved and their roles and functions in developing sustainable tourism in the research area can be explained. Some of the things that were found in the research concerning the presence of various parties in the development of sustainable tourism in the research area can be explained as follows.

1.  Government as Regulator

In this context, the government includes the village, district, provincial, and central governments. Each level of government has its authority as mandated by laws and regulations concerning regional government. The government's role as a regulator is to prepare guidance to balance the implementation of development through the issuance of the necessary rules, both in the framework of structuring existing tourist destinations and in the context of regulating various aspects of institutional tourism governance. In general, the government has the authority to issue multiple fiscal policies that can incentivize specific

tourist destinations to develop faster, for example, by providing subsidies to airlines to pilot flights to newly opened tourist areas. As another example, the government can provide support or assistance to travel agents to lure tourists to visit, especially tourists from foreign countries.

2.    Government as Facilitator

Aside from being a regulator, the government can also act as a facilitator by providing various facilities needed in structuring and developing a tourist destination. The findings of this research should be a concern of the government. The government should facilitate various limitations in the research area so that the potential of the tourism framework can be immediately operationalized. As a relatively new area in tourism, the research area has various supporting facilities that are inadequate, such as public transportation, good parking lots, and other public facilities. In human resource development, the government has the capacity and authority to improve the ability and quality of tourism managers, both at the managerial level and at the level of executors and supervisors in the field. In optimizing the government's role as a facilitator, local governments also need to encourage the participation of other parties or competent stakeholders in developing tourism facilities and infrastructure. The government must also promote coordination and cooperation between stakeholders, such as the private sector, universities, civil society (NGOs), and the media.

3.    Involvement of the Private Sector

The involvement of the private sector in tourism development, including sustainable tourism, will accelerate the development process. This involvement is in accordance with the principles of sustainable tourism. Tourism development should be able to create jobs and increase people's incomes. As investors, elements of the private sector have potential that can be utilized economically to carry out various activities for structuring destinations and managing tourist areas with the government and the community. However, the principles of ownership by the community and activities driven by the local community should still be promoted.

From research conducted by Elmo et al. [57], it was concluded that the private sector could play a role in developing sustainable tourism as long as it can contribute and or gain economic benefits. Besides that, the aspect of corporate resilience, both economically and socially as well as environmentally, is also a determining factor in whether a private company can grow.

4.    Higher Education Institution Involvement

Sustainable tourism development is science-based development. Therefore, in implementing sustainable tourism, the role of universities is vital. Universities can collaborate with the local community from the initial phases of sustainable tourism development, such as exploring potential, planning development, and even identifying development models that can be implemented. In addition, lecturers and students can be directly involved in related learning programs on campus.

The role of higher education institutions is oriented more toward developing and establishing the direction of sustainable tourism. As researched by Qian et al. [58], six significant themes need attention in developing sustainable tourism: climate change, behavioral studies, poverty reduction, volunteer tourism, policy instruments, and indigenous tourism. Here, the role of universities is expected to contribute.

5.    Involvement of Civil Society or NGOs

Today the existence and role of civil society in development, including tourism, is essential. This includes non-governmental organizations that act as a liaison between the community, government, and outside agencies concerned with sustainable tourism development. Several non-governmental organizations have emerged in the research area, including Forest Farmers Groups, Karang Taruna (Youth Organization), Tourism Awareness

Community Groups (Pokdarwis), and others. From the results of this study, it was found that the Forest Farmers Group (KTH) in the research area plays a reasonably active role in developing tourism. KTH already has a management structure and is quite active in the field, especially in providing security for the parked vehicles of visiting tourists.

According to several researchers [59,60], the role of civil society or non-governmental organizations is very strategic in developing sustainable tourism. The role of these NGOs is not limited to the planning and development process; it also extends to integrating tourism development with rural development and society in general.

6.    Media Engagement

Finally, media involvement has a strategic role, especially in marketing or promoting sustainable tourism. With this media involvement, the wider community will quickly recognize a new tourist destination area. Furthermore, the media can also function in the context of the education process for the public and tourists. As explained earlier, one of the principles of sustainable tourism is continuous education, both for the community and visitors, regarding the importance of preserving existing natural and cultural resources. Thus, through the media, various educational processes can be carried out.

Chatterjee and Dsilva [61] examined the critical role of the media in sustainable tourism development. It was concluded from their research that there is a very close relationship between media and sustainable tourism, especially in the context of developing relatively underdeveloped tourist areas. The study also concluded that the media have a role in marketing sustainable tourism areas targeting those who like to travel, including groups of parents who are already literate on the internet or social media.

## 4. Discussion

In relation to the level of readiness of the community in the context of implementing sustainable tourism, it can be concluded that the community is in the "very ready" category. This conclusion means the community was prepared in terms of the various forms of capital examined in this study.

First is economic capital. Tourism development requires heavy investment. Many things must be addressed in the context of preparing for tourism development. One of the most important factors is the preparation of the infrastructure needed by tourism actors, starting from arranging destinations. These facilities will include tourist attractions, culinary centers, trash bins, and adequate toilets. As a sustainable tourist destination, under the principles of sustainable tourism, which requires an educational process for visitors, one of the necessary elements of infrastructure will be an environmental education room. It will be a vehicle for education, for both local community members and visitors, on the importance of preserving the environment and maintaining the existing cultural heritage. Besides that, it should also be ensured that the local community enjoys the economic benefits of sustainable tourism. Therefore, preparing local communities to engage and play an active role in sustainable tourism development needs special attention. Training and social engineering should be carried out measurably so that people do not become "spectators" in their tourist areas. To ensure sufficient economic capital, sustainable tourism actors can invite outside investors to finance various infrastructure needs. However, the invited investors should understand well the principles of sustainable tourism development [57]. Apart from investors, tourism actors can also ask the government, in this case as a facilitator, to assist in procuring various elements of necessary tourism infrastructure [1]. As with the private sector, the government should also remain guided by the principles of sustainable tourism.

Second, in terms of cultural capital, every area where tourism develops must have artistic potential hat be used as an attraction. Culturally based tourism is included in sustainable tourism. In this study, what is meant by cultural capital is related to educational issues for the local community. The assumption is that if the awareness and education level of the local community are high, then it is certain that the implementation of the principles of tourism development will work well. In this case, the aspect of human resources is

a crucial issue. Therefore, in any action to develop sustainable tourist destinations, this human resource issue should be a significant concern. The social engineering process in preparing each new destination must be carried out as effectively as possible. It is essential not to repeat the past mistakes of other regions or countries [19].

In this study, the role of education in sustainable tourism development was specifically reviewed. Education, both in the context of formal and informal education, has a vital role in developing sustainable tourism. According to Idajati et al. [54], the intended education is education for visitors and education for the local community. Education, in this case, is related to efforts to increase public awareness of the importance of preserving natural resources, which is one of the attractions in the world of tourism that must be maintained.

Third, social capital plays a very strategic role in sustainable tourism development. This role is based on the principles of sustainable tourism, where participation and ownership by the local community are the main factors. By assuming ownership of the existing social capital, members of the local community will be able to actively participate in developing tourism in their area. By exploiting cultural values that grow and develop in their area, as one example, they can make these traditional cultural values a unique attraction in sustainable tourism. Likewise, according to Zach and Racherla, utilizing social networks owned by the community will enable tourist destinations to develop quickly [29]. Through its networks, the community can also build cooperation with fellow tourism actors in other areas with the principle of mutual benefit. Moreover, the issue of sustainable tourism is a global issue, so communication with actors in other countries is also very open.

Fourth is symbolic capital. In the context of this study, symbolic capital relates to leadership, honor, and fame. In implementing sustainable tourism, symbolic capital plays a significant role. The leadership factor in society has a strategic position. Effective leadership will be the key to successfully implementing programs, including those in the field of tourism. In relation to the level of community readiness in implementing sustainable tourism, this aspect of symbolic capital is in the "high" category. The implication is that various processes in the field will be significantly influenced by the presence of a leader who can mobilize the community [6]. This leader presence is also based on the principles of sustainable tourism, where leadership issues are included.

Furthermore, several conclusions can be drawn regarding factors influencing community readiness to develop sustainable tourism. First, from in-depth interviews conducted with several key figures in the research area, it could be seen that from the economic and socio-cultural aspects, the community was ready to organize sustainable tourism. Secondly, from the observations, the community had begun to implement tourism activities at a certain level. Such activity can indicate community readiness to carry out highly sustainable tourism activities. Implementation of the principles of sustainable tourism has also begun, such as entrepreneurial activities that can encourage job creation. With the start of tourism activities in the research area, Pink Beach, as one of Lombok Island's leading tourism destinations, can be an initial capital with great potential for development.

For this reason, government intervention and private sector involvement are factors that will significantly determine the direction of tourism development in the research area [1]. The penta-helix approach, in which elements of government, the private sector, universities, NGOs, and media are actively involved, is expected to be effective in accelerating tourism development in this area. In tourism development in the third world, including Indonesia, the government's role is still very much needed. The role of the government is not only that of a regulator but also as a facilitator. In various tourist areas, the government assists in the development of much-needed infrastructure and transportation facilities.

One model developed entirely within the world of tourism includes a role for private investors in developing and managing tourist destinations. The management of tourism by the private sector tends to be commercial because the character of the private sector is profit-oriented [15]. However, through understanding and applying the principles of sustainable tourism, aspects of tourism commercialization that tend to damage the environment can be averted so that concerns about environmental damage and the non-empowerment of

local communities can be prevented. In this context, the presence of higher education institutions as part of the penta-helix approach becomes essential. The role of higher education in the development of sustainable tourism is to produce the various technologies and innovations needed for the preservation of natural resources and the development of human resources [5]. The role of higher education becomes more effective when side-by-side with the presence of civil society or non-governmental organizations and the world of media. The presence of non-governmental organizations and the media can act as a counterweight or a kind of control agency over the activities of other penta-helix elements. They can function as strategic partners that can make the penta-helix model effective in implementing sustainable tourism. They can also act as a bridge of interest between the community and other tourism actors. The synergy between sustainable tourism actors will guarantee the presence of sustainable tourism, which can improve the welfare of the surrounding community.

**5. Conclusions**

From the discussion above, the following conclusions can be drawn. First, the people in the research area are ready to develop sustainable tourism. Their level of readiness can be described from various aspects of capital, namely economic capital, whereas the level of community readiness is included in the "very high" category. In addition, the availability of adequate facilities and infrastructure indicates this category. Regarding cultural capital, the level of community readiness is in the "high" category. This category means that the community's knowledge is high but still relatively low regarding skills. The community's level of readiness in terms of the social capital is also very high, as indicated by several indicators such as social values/norms, social relations, and the trust held by the community. In terms of symbolic capital, the community's readiness level is also high, which means that the leadership factor in the research area is quite effective in moving the community toward sustainable tourism development.

Second, it can be concluded that the aspects that influence sustainable tourism development are economic, social, and cultural. From an economic perspective, sustainable tourism can add jobs for the community and simultaneously increase residents' incomes. Regarding the social aspect, formal and informal education both play a very strategic role. From the cultural aspect, implementing sustainable tourism also positively impacts maintaining the cultural values embraced by the community. Furthermore, the implementation of the principles of sustainable tourism is likely to increase public awareness of the importance of preserving natural resources.

Third, the stakeholders who play a role in developing sustainable tourism in the research area are the government, the private sector, universities, non-governmental organizations, and the media. The government plays a role in preparing regulations and providing various necessary tourism facilities and infrastructure. The private sector parties act as investors and, together with the community, manage tourist destinations while still applying the principles of sustainable tourism. Universities play a role in the study, preparation, and development of tourist destinations so that the possibility of negative impacts can be avoided. NGOs play a role in the process of educating the public and providing assistance to managers in implementing good tourism destination management. Finally, the media help inform the public and visitors and assist in marketing existing tourist destinations. The five elements of stakeholders are described as a penta helix.

**Funding:** This research received no external funding.

**Data Availability Statement:** The author confirms that the data supporting the findings of this study are available within the article and its permission for using Table 1.

**Acknowledgments:** I express my gratitude and highest appreciation to my colleague M. Zaenul Muttaqin from Cendrawasih University, who has helped me prepare this manuscript according to the template from the available journal. Additionally, I thank my students, Isnan, Kurnia, Mimin, Elina, Henny, Dina, and Dian, who have helped me carry out this research.

**Conflicts of Interest:** The author declares no conflict of interest.

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
