# Peer review of "Community Readiness in Implementing Sustainable Tourism on Small Islands: Evidence from Lombok, Indonesia"

_sustainability, doi:10.3390/su15129725_

Round 1

Reviewer 1 Report

Abstract

It is not customary to include numbers for enumeration in the abstract. It should be written without numbers.

1. Introduction

Insert a map of the area being studied to make it clearer to the readers.

Line 80 - Remove the site from the text and put it in the references

Line 143

For abbreviations, the first time they are mentioned, give the full name

Line 180-211

It is not necessary to specify the data collection with separate numbers and put special subheadings. Those concepts should only be explained by sentences.

Line 214

Why ‘quota sampling’ is written twice?

Line  220

My opinion is that sample size of 60 people is not enough for in dept research. I can understand that tourism is at the beginning, so the author should have bigger sample in future research.

2. Materials and Methods - The Materials and Methods chapter must be revised to make it more understandable.

Line  241,

Table No 2. Is not understandable. Why this table is not translated in to English?

Line 297

In the following text, the authors refer to the results from table 2., which has not been translated, so a review cannot be performed.

3. Results

Table No 3. Is not understandable. Why this table is not translated in to English?

Influential aspects / From line 508 – 618

I don't think it is necessary to describe the conversations in this way. The author should state his conclusions based on the conducted interviews

References

In some references, the years are not specified and this should be corrected.

Despite the desire to do a quality review, this is not possible in the case of this paper. The reason is that some of the tables have not been translated into English and therefore, the research and the obtained results cannot be followed.

I think that first the authors should make those revisions in the text so that a review can be done.

Author Response

Dear Reviewer 1,

Abstract

It is not customary to include numbers for enumeration in

I have changed it already.

  1. Introduction

Insert a map of the area being studied to make it clearer to the readers.

Ok, I will insert it later in the final version of the manuscript.

Line 80 - Remove the site from the text and put it in the reference

I have removed it already.

Line 143

For abbreviations, the first time they are mentioned, give the full name

I have changed it by giving the full name.

Line 180-211

It is not necessary to specify the data collection with separate numbers and put special subheadings. Those concepts should only be explained by sentences.

I have changed the paragraph’s structure.

Line 214

Why ‘quota sampling’ is written twice?

I have deleted one of them.

Line 220

My opinion is that sample size of 60 people is not enough

for in dept research. I can understand that tourism is at the beginning, so the author should have bigger sample in

future research.

  1. Materials and Methods - The Materials and Methods chapter must be revised to make it more

Line 241,

Table No 2. Is not understandable. Why this table is not translated in to English?

I am sorry, but I have translated it into English already.

Line 297 

In the following text, the authors refer to the results from

table 2., which has not been translated, so a review cannot be performed.

I am sorry, but I have translated it into English already.

  1. Results

Table No 3. Is not understandable. Why this table is not translated in to English?

I am sorry, but I have translated it into English already.

Influential aspects / From line 508 – 618

I don't think it is necessary to describe the conversations in this way. The author should state his conclusions based on the conducted interviews

I think it deals with academic writing style in my university. I am considering to change it if it relates to journal style.

References

In some references, the years are not specified and this should be corrected.

   I have corrected them already.

Thank you very much for your valuable suggestion to improve the quality of my manuscript. Enclosed is the revision that I have made so far. I look forward to your response. I will be happy to revise it again as per your suggestion. 

Best regards,

Rosiady Sayuti

Reviewer 2 Report

The author should define what is the concept of "readiness", The indicators are listed but not explained (For example, infrastructures, what are we talking about in this context? Airports and roads or recycling plants? )

Regarding the questionnaire, it is not clear what the choice of the sample is (local terms are used which should be explained in some way), nor are the questions, it talks about yes or no answers, but what are the questions? This is a determining factor in order to be able to analyse the answers.

In general I think that the methodology, especially the quantitative one, should be revised to make it more detailed and understandable.

There are many enumerations that are sometimes nested, leaving an unclear structure. I believe that this can be rectified without difficulty.

Some of the tables are not in English (e.g. table 4).

Some redundant paragraphs, like;

"1) Observation is the activity of observing directly to observe the activities carried out by the research object closely. Observations made by researchers collected data from the field by looking directly at the condition of the people in the research area."

It must be rewrite.

Author Response

Dear Reviewer 2,

The author should define what is the concept of "readiness", The indicators are listed but not explained (For example,

infrastructures, what are we talking about in this context? Airports and roads or recycling plants? )

I have added the concept’s explanation of readiness in lines 101-103.

Regarding the questionnaire, it is not clear what the choice of the sample is (local terms are used which should be explained in some way), nor are the questions, it talks about yes or no answers, but what are the questions? This is a determining factor in order to be able to analyse the answers.

I have added the explanation in lines 255 to 259.

In general I think that the methodology, especially the quantitative one, should be revised to make it more detailed and understandable.

There are many enumerations that are sometimes nested,

leaving an unclear structure. I believe that this can be rectified without difficulty.

Ok, I  have corrected  already. Thank you for the suggestions

Thank you very much for your valuable suggestion to improve the quality of my manuscript. Enclosed is the revision that I have made so far. I look forward to your response. I will be happy to revise it again as per your suggestion. 

Best regards,

Rosiady Sayuti

Reviewer 3 Report

Recommendations

The Abstract and the Conclusion include the statement that "Aspects that influence the development of sustainable tourism are economic, social, and cultural." (Lines 19, and 814-815), missing the Natural aspect, while its importance is underlined in the same article for many times (see lines 40, 42, 49, 65, 92, 100, 103, 109, 139, 349, 614, 698, 785). I would recommend either including this component in the article or emphasizing that the research is focused on economic, social, and cultural aspects.

The text contains several technical shortfalls:

The repeated expression: "Quota sampling or quota sampling. (Line 214); 

The brackets contain perplexing signs "(>>>>>>>)" instead of reference symbols. (Line 405);

The tables and their subtitles contain inappropriate letters: Persentase (Table 2), Kriteria (Table 2, 4), Karakteristik Respondent (Table 3), indikator (table 4) 

The text contains mismatches and/or needs clarifications:

Line 326: ".., as shown in Table 2 above,…". To my understanding, the referred table should be 4 (not 2);

Line 344:  contains - "…a rate of 77%.", that is not viewed in the relevant table;

Line 387-388: states - "…, it is shown that 43.3% of respondents have work experience in the tourism sector." while Table 4 shows – 14%;

Line 393: marks – "…8.3% of the respondents could speak English, and 35% had…..", the sources are unclear;

Lines 421 and 425: contain the indicator 95% referring to Table 2 where there is no such indicator

Line 445: states – mentions the rate of visiting neighbors (homes) as "78%", while in the table it is 83%

Line 453:  the involvement in promoting tourism is mentioned at the rate of 55%, in the relevant table it is - 43%

Line 463: regarding the cooperation with outsiders the stated rate is 85% (in the text), 79% (in the relevant table)

Line 486:  the same mismatch – 86.7% in the text, in oppose to 59% in the table,

Line 490:  the same disparity – 91.7% in the text, and 57% in the table;

The number of stated sources in the References does not include the dates of their edition/accesses

Suggestions:

ü  The Table 3, 3-rd column is titled as "%", I would recommend clarifying - % of what?

ü  Table 1, and 4 contains the abbreviation HP, which is recommended to clarify when it is first mentioned (usually it means Hire Purchase)

ü  Education is considered as part of the Cultural aspect, I would move to the Social one.

Author Response

Dear Reviewer 3,

Recommendations

The Abstract and the Conclusion include the statement that "Aspects that influence the development of sustainable tourism are economic, social, and cultural." (Lines 19, and 814-815), missing the Natural aspect, while its importance is underlined in the same article for many times (see lines 40, 42, 49, 65, 92, 100, 103, 109, 139, 349, 614, 698, 785). I would recommend either including this component in the article or emphasizing that the research is focused on economic, social, and cultural aspects.

I have added the natural aspect in line 20. Thank you

The text contains several technical shortfalls:

The repeated expression: "Quota sampling or quota sampling. (Line 214);

I have deleted one of them.

The brackets contain perplexing signs "(>>>>>>>)" instead of reference symbols. (Line 405);

I have corrected it already.

The tables and their subtitles contain inappropriate letters: Persentase (Table 2), Kriteria (Table 2, 4), Karakteristik Respondent (Table 3), indikator (table 4)

I am sorry, but I have corrected it already.

The text contains mismatches and/or needs clarifications:

Line 326: ".., as shown in Table 2 above,…". To my understanding, the referred table should be 4 (not 2);

I have corrected it already.

Line 344: contains - "…a rate of 77%.", that is not viewed in the relevant table;

I have corrected it already.

Line 387-388: states - "…, it is shown that 43.3% of respondents have work experience in the tourism sector." while Table 4 shows – 14%;

I have corrected it already.

Line 393: marks – "…8.3% of the respondents could speak English, and 35% had…..", the sources are unclear;

I have corrected it already.

Lines 421 and 425: contain the indicator 95% referring to Table 2 where there is no such indicator

I have corrected it already.

Line 445: states – mentions the rate of visiting neighbors (homes) as "78%", while in the table it is 83%

I have corrected it already.

Line 453: the involvement in promoting tourism is mentioned at the rate of 55%, in the relevant table it is - 43%

Line 463: regarding the cooperation with outsiders the stated rate is 85% (in the text), 79% (in the relevant table)

Line 486: the same mismatch – 86.7% in the text, in oppose to 59% in the table,

Line 490: the same disparity – 91.7% in the text, and 57% in the table;

The number of stated sources in the References does not include the dates of their edition/accesses

I have corrected it already.

Suggestions:

ü  The Table 3, 3-rd column is titled as "%", I would recommend clarifying - % of what?

I have corrected it already. I changed “%” to “percentage”

ü  Table 1, and 4 contains the abbreviation HP, which is recommended to clarify when it is first mentioned (usually it means Hire Purchase)

I have corrected it already. HP means Mobile Phone

ü  Education is considered as part of the Cultural aspect, I would move to the Social one.

In this study, education is part of cultural aspect.

Thank you very much for your valuable suggestion to improve the quality of my manuscript. Enclosed is the revision that I have made so far. I look forward to your response. I will be happy to revise it again as per your suggestion. 

Best regards,

Rosiady Sayuti

Reviewer 4 Report

It is an interesting study; however, there is room for improvement as suggested below : 

1. Introduction can be related to SDGs and what policies and strategies have been implemented in Indonesia to sustain the industry 

2. The justification of the study conducted in the chosen place needs to be strengthened. What is the issue at Lombak in sustaining tourism and why CBT needed? 

3. 60 respondents for quantitative is not acceptable. 

4. Some of the findings presented in Malay- like demographic information

5. References do not include the year of publication. How to identify the latest journals? 

6. Need more comprehensive and constructive arguments in analysis, discussion and conclusion

7. Abstract should have the limitation and recommendations for future study. 

Please proofread the article before submit. 

Author Response

Dear Reviewers 4,

  1. Introduction can be related to SDGs and what policies and strategies have been implemented in Indonesia to sustain the industry

I have added in lines 38-44.

  1. The justification of the study conducted in the chosen place needs to be strengthened. What is the issue at Lombok in sustaining tourism and why CBT needed?

I have explained it in lines 80-96

  1. 60 respondents for quantitative is not

According to some references (Sugiyono, 2013; ) the number of appropriate samples for the correlational study is in the range of 30 to 500.

  1. Some of the findings presented in Malay- like demographic information.

I am sorry, but I have translated it into English already.

  1. References do not include the year of How to identify the latest journals?

I have corrected it already.

  1. Need more comprehensive and constructive arguments in analysis, discussion and conclusion

OK, I have done it through this reviewing process

  1. Abstract should have the limitation and recommendations for future

The limitation of the study and recommendations for future research are optional. But I will create them later, as you suggested. Thank you.

Thank you very much for your valuable suggestion to improve the quality of my manuscript. Enclosed is the revision that I have made so far. I look forward to your response. I will be happy to revise it again as per your suggestion. 

Best regards,

Rosiady Sayuti

Reviewer 5 Report

It was with great interest that I read the article “Community Readiness in Implementing Sustainable Tourism on Small Islands: Evidence from Lombok, Indonesia”. The subject of the study is actual and very pertinent, when the discussion about sustainability, particularly in the scope of tourism activity, gains an expression never seen before, as well as the role played by local communities in promoting sustainable tourism.

Taking in account the current version of the article, I believe some improvements are needed. I hope that the following suggestions can serve as a guide for the authors to revise and to improve the final manuscript.

1. Abstract: I miss I little bit of introduction/background before the purpose. About the methodology it should be clarified which methods, quantitative and qualitative, were used. One of the objectives present here is “3) Knowing the role of stakeholders in sustainable tourism development” But in the last paragraph only identifies them without mentioning their role as described in the goal. It would be interesting to add the practical implications of this study.

2. Key-words:  I think it would make sense to add “stakeholders' role”.

3. Introduction section: the bibliographical references are not in accordance with the journal's rules. This should be revised throughout the all text (e.g. [1, 2] and not [1], [2]; According to Guizzardi et al., [9] and not According to [9]. Line 56 “According to several authors” - who are these authors? They are not mentioned! It should have explored a little more the tourist dimension of Lombok Island, that is, presented more concrete data. The site on lines 80 and 81 is unnecessary. The authors do not explain why these three particular villages were chosen.

4. Literature Review section: must be individualized from the introduction. It is too succinct and should be expanded. The bibliography about community-based tourism development is very extensive.

5. Materials and Methods section: I found this section quite confusing; it needs clarification and organization. It should be more objective and clearly show which methodologies are applied and on whom. For example, the questionnaire was applied to whom? How many? when? how? The same with the interview. The key informants were the stakeholders? If the information about the research sample has to do with the questionnaire, why does it appear in its own section after you have explained who the informants were in the interview part? The table 1 has to do with questionnaires, right?

6. Results section: too extensive and too descriptive part. The bibliographic references do not appear correctly. The text in the “The Role of Stakeholders in tourism development” point results from what? From the interviews? From literature review?

7. Discussion section: too descriptive. From my point of view there´s a lack of a concrete discussion. The dialogue between the findings and previous research does not exist.

8. Conclusion section: is poorly written. I recommend the authors to summarize the main contribution of this research, explain the practical implications and the main limitations of the research as well as future research directions.

9. In the reference list some information must be reviewed because some references do not follow the rules of the journal (https://www.mdpi.com/authors/references). The list of references should be improved.

10. From my point of view English must be revised by a professional.

From my point of view English must be revised by a professional.

Author Response

Dear Reviewers 5,

  1. Abstract: I miss I little bit of introduction/background before the purpose. About the methodology it should be clarified which methods, quantitative and qualitative, were used. One of the objectives present here is “3) Knowing the role of stakeholders in sustainable tourism development” But in the last paragraph only identifies them without mentioning their role as described in the goal. It would be interesting to add the practical implications of this study.

    I have added the introduction in the abstract, lines 8-10.

    This study apply mixed methods; to explain the objective number 1, we used quantitative methods, while to explaine the objective number 2 and 3, we apllied qualitative methods
  2. Key-words: I think it would make sense to add “stakeholders' role”.

    I have added it.

  3. Introduction section: the bibliographical references are not in accordance with the journal's rules. This should be revised throughout the all text (e.g. [1, 2] and not [1], [2]; According to Guizzardi et al., [9] and not According to [9]. Line 56 “According to several authors” - who are these authors? They are not mentioned! It should have explored a little more the tourist dimension of Lombok Island, that is, presented more concrete data. The site on lines 80 and 81 is unnecessary. The authors do not explain why these three particular villages were chosen.

    I have fixed them already.

  4. Literature Review section: must be individualized from the introduction. It is too succinct and should be expanded. The bibliography about community-based tourism development is very extensive.

    I have improved it to some extent.

  5. Materials and Methods section: I found this section quite confusing; it needs clarification and organization. It should be more objective and clearly show which methodologies are applied and on whom. For example, the questionnaire was applied to whom? How many? when? how? The same with the interview. The key informants were the stakeholders? If the information about the research sample has to do with the questionnaire, why does it appear in its own section after you have explained who the informants were in the interview part? The table 1 has to do with questionnaires, right?

    I have explained about the questionnaire, respondent, when and how the data was collected in lines 210-218. Explanation about the key informant for qualitative data can be read in lines 234-236. Yes, the Table 1 has to do with the questionnaire.

  6. Results section: too extensive and too descriptive part. The bibliographic references do not appear correctly. The text in the “The Role of Stakeholders in tourism development” point results from what? From the interviews? From literature review?
    The bibliographic references, to some extent, have been fixed. “The Role of Stakeholders in tourism development” based on the interviews and literature review
  7. Discussion section: too descriptive. From my point of view there´s a lack of a concrete discussion. The dialogue between the findings and previous research does not exist.

    I have added some paragraphs to improve the discussion.

  8. Conclusion section: is poorly written. I recommend the authors to summarize the main contribution of this research, explain the practical implications and the main limitations of the research as well as future research directions.

    I have improved it already. You may check them in lines 896-902 and 908-916
  9. In the reference list some information must be reviewed because some references do not follow the rules of the journal (https://www.mdpi.com/authors/references). The list of references should be improved.

    I have fixed it already.

  10. From my point of view English must be revised by a professional.

    I will proofread it again before submitting the final version of this manuscript.

Thank you very much for your valuable suggestion to improve the quality of my manuscript. Enclosed is the revision that I have made so far. I look forward to your response. I will be happy to revise it again as per your suggestion. 

Best regards,

Rosiady Sayuti

Round 2

Reviewer 1 Report

The paper should be done according to the requirements of the magazine

Given that it is about the beginning of the development of tourism, I understand that the author/s could not have more respondents.

No dot is needed after the title of Table 2

Line - 532/642

Written like this, it burdens the text, which loses its quality. The author should state conclusions based on the conducted interviews.

Check the References and whether everything is entered from text (e.g. UN reference)?

Author Response

Dear Reviewer #1,

Thank you for your response. I have revised the manuscript as you suggested. My response is as follows:

The paper should be done according to the requirements of the magazine.

Ok

Given that it is about the beginning of tourism development, I understand that the author/s could not have more respondents.

Thank you for your understanding.

No dot is needed after the title of Table 2

Ok, I deleted it already

Line - 532/642

Written like this, it burdens the text, which loses its quality. The author should state conclusions based on the conducted interviews.

I revised them already, as you suggested.

Check the References and whether everything is entered from the text (e.g., UN reference).

Ok, I checked it. Thank you.

The manuscript is attached. 

Best regards,

Rosiady Sayuti

Reviewer 3 Report

After implemented corrections the publication is acceptable.

Author Response

Dear Reviewer 3,

Thank you for taking the time to review my manuscript. I appreciate it. I hope my manuscript will be published in the journal soon. Enclosed is the final version of my manuscript. 

Thank you and best regards,

Rosiady Sayuti

Reviewer 4 Report

Thank you for improvement made.

Thank you for improvement made.can proceed for publication 

Author Response

Dear Reviewer 4,

Thank you for taking the time to review my manuscript. I appreciate it. I hope my manuscript will be published in the journal soon. Enclosed is the final version of my manuscript. 

Thank you and best regards,

Rosiady Sayuti

Reviewer 5 Report

Thank you for submitting the revised version of your paper. I appreciate the work of the Author and changes introduced to the manuscript, which substantially improved it. Congratulations on that. However, some of the points of the first review were ignored, so I would suggest you re-read the comments from the first revision. Regarding this new version presented I would point out the following:

-        The authors still do not explain why these three particular villages were chosen.

-        Literature Review had an improvement but not enough from my point of view.

-        In discussion section the dialogue between the findings and previous research is still missing.

-        The bibliographical references should be revised through the text and in the final list because some of them are not in accordance with the journal's rules.

-        Attention should be paid to the formatting of the text.

From my point of view English must be revised by a professional.

Author Response

Dear Reviewer 5,

Thank you for your critical point of view regarding my manuscript. It will increase the quality of this manuscript and make it more interesting for the reader in academic circumstances. Here are my responses: 

Thank you for submitting the revised version of your paper. I appreciate the author's work and the changes introduced to the manuscript, which substantially improved it. Congratulations on that. However, some of the points of the first review were ignored, so I suggest you re-read the comments from the first revision. Regarding this new version presented, I would point out the following:

Author response: I have read the first review and have made some corrections, such as adding the abstract in the last sentence regarding the use of the result of this study to create government policy. Also, as you suggested, I put some references in the discussion section. I also added the manuscript with an education discussion, as indicated by the editor.

-        The authors still do not explain why these three particular villages were chosen.

Author response: I have explained already in lines 232-235.

-        Literature Review improved but not enough from my point of view.

Author response: I have improved it, to some extent.

-        In the discussion section, the dialogue between the findings and previous research is still missing.

      Author response: I have put some references in the discussion section to create a dialog with previous studies.

-        The bibliographical references should be revised through the text and in the final list because some of them are not in accordance with the journal's rules.

      Author response: I have fixed them already

-        Attention should be paid to the formatting of the text.

OK, thank you.

I look forward to your response.

Thank you and best regards,

Rosiady Sayuti

Round 3

Reviewer 5 Report

Thank you for the revised version submited.

In the first paragraph of the introduction remain referencing issues that do not conform to the journal's rules (pp. 36, 42, 48, 58).

Revise the formatting of the text.

Author Response

Dear Reviewer 5,

Thank you for suggesting improving my manuscript quality and meeting the journal style. I have revised them already. I attached the manuscript along with this letter. 

Thank you and best regards,

Rosiady Sayuti
